# Synthesis and In Vitro Evaluation of Gold Nanoparticles Functionalized with Thiol Ligands for Robust Radiolabeling with ^99m^Tc

**DOI:** 10.3390/nano11092406

**Published:** 2021-09-15

**Authors:** Adamantia Apostolopoulou, Aristeidis Chiotellis, Evangelia-Alexandra Salvanou, Konstantina Makrypidi, Charalampos Tsoukalas, Fotis Kapiris, Maria Paravatou-Petsotas, Minas Papadopoulos, Ioannis C. Pirmettis, Przemysław Koźmiński, Penelope Bouziotis

**Affiliations:** 1National Center for Scientific Research “Demokritos”, Institute of Nuclear & Radiological Sciences & Technology, Energy & Safety, Agia Paraskevi, 15341 Athens, Greece; madoapostol@gmail.com (A.A.); achiotel@rrp.demokritos.gr (A.C.); salvanou@rrp.demokritos.gr (E.-A.S.); kmakripidi@gmail.com (K.M.); ctsoukal@rrp.demokritos.gr (C.T.); fotiskp@gmail.com (F.K.); mparavatou@rrp.demokritos.gr (M.P.-P.); mspap@rrp.demokritos.gr (M.P.); ipirme@rrp.demokritos.gr (I.C.P.); 2Department of Pharmacy, National and Kapodistrian University of Athens, Panepistimiopolis Zographou, 15771 Athens, Greece; 3Centre of Radiochemistry and Nuclear Chemistry, Institute of Nuclear Chemistry and Technology, Dorodna 16 Str., 03-195 Warsaw, Poland; p.kozminski@ichtj.waw.pl

**Keywords:** gold nanoparticles, ^99m^Tc-carbonyls, radiolabeling, cytotoxicity, MTT, hemolysis assay

## Abstract

Radiolabeled gold nanoparticles (AuNPs) have been widely used for cancer diagnosis and therapy over recent decades. In this study, we focused on the development and in vitro evaluation of four new Au nanoconjugates radiolabeled with technetium-99m (^99m^Tc) via thiol-bearing ligands attached to the NP surface. More specifically, AuNPs of two different sizes (2 nm and 20 nm, referred to as Au^(2)^ and Au^(20)^, respectively) were functionalized with two bifunctional thiol ligands (referred to as L_1_H and L_2_H). The shape, size, and morphology of both bare and ligand-bearing AuNPs were characterized by transmission electron microscopy (TEM) and dynamic light scattering (DLS) techniques. In vitro cytotoxicity was assessed in 4T1 murine mammary cancer cells. The AuNPs were successfully radiolabeled with ^99m^Tc-carbonyls at high radiochemical purity (>95%) and showed excellent in vitro stability in competition studies with cysteine and histidine. Moreover, lipophilicity studies were performed in order to determine the lipophilicity of the radiolabeled conjugates, while a hemolysis assay was performed to investigate the biocompatibility of the bare and functionalized AuNPs. We have shown that the functionalized AuNPs developed in this study lead to stable radiolabeled nanoconstructs with the potential to be applied in multimodality imaging or for in vivo tracking of drug-carrying AuNPs.

## 1. Introduction

Nanoparticles are materials with dimensions at the nanometer scale (1–100 nm) and over the last decades have been widely used in theranostic applications, thus playing an effective role in the diagnosis and therapy of cancer by offering multiple possibilities in oncology [1]. The aim of nanotechnology is to allow efficient accumulation of the nanodrug in the target organ or tissue, regardless of the method and the route of administration, while limiting unwanted toxic effects due to the administration of high concentrations of conventional anticancer drugs.

In particular, gold nanoparticles (AuNPs) have been extensively investigated for the development of dual-modality imaging agents, as well as for theranostic applications due to their unique physicochemical properties, ease of surface functionalization with different chemical entities, low toxicity, and biocompatibility. They also present high sensitivity and can be detected even at low concentrations [1,2,3]. Gold nanoparticles are synthesized by various methods, including the reduction of gold ions (Au^3+^) to produce gold atoms (Au^0^) by using reducing agents such as amino acids, UV light, or citrate, but also eco-friendly synthetic processes in aqueous solutions rather than organic solvents, where ascorbic acid (AA) has been used instead of traditional reductants [4,5,6]. The use of AuNPs in diagnostic applications is based on the passive targeting of the tumor site via the enhanced permeability and retention (EPR) effect, where the nanoparticles, due to their small size, extravasate into tumors through leaky vasculatures, where they are retained due to poor lymphatic drainage. AuNPs can also reach the disease site via active targeting, where targeting ligands (such as peptides or monoclonal antibodies) attached to the nanoparticle surface recognize specific receptors overexpressed on the surface of tumor cells, resulting in enhanced accumulation in the targeted organ [7,8].

In order to investigate the in vivo kinetics of AuNPs, these can be radiolabeled with a wide variety of radionuclides. However, NP radiolabeling is associated with some significant disadvantages. For example, some radiolabeling modifications have been shown to affect the physicochemical properties of the nanoparticles. Moreover, AuNPs present many shapes, sizes, and morphologies, leading to a huge variety of radiolabeled compounds. Most importantly, targeting efficiency of the radiolabeled NPs is lost if the radiolabel does not remain associated with the NPs for a sufficient period of time [9].

For diagnosis, short-lived gamma- or positron-emitting radionuclides, including technetium-99m (^99m^Tc), gallium-68 (^68^Ga), iodine-123 (^123^I), indium-111 (^111^In), fluorine-18 (^18^F), and copper-64 (^64^Cu), are extensively used for radiolabeling AuNPs [9,10,11,12,13,14,15,16,17,18]. ^99m^Tc is the most commonly used radionuclide in diagnostic imaging, due to its ideal nuclear characteristics, such as emission of low-energy γ-rays (140 keV) and suitable half-life (6.02 h), as well as its convenient availability from ^99^Mo/^99m^Tc generators [19,20]. Labeling via the [^99m^Tc][Tc(H_2_O)_3_(CO)_3_]^+^ carbonyl core is a very efficient way of radiolabeling with ^99m^Tc and was first introduced by Alberto et al. in 1998 [21].

AuNPs can also be radiolabeled with therapeutic radioisotopes such as actinium-225 (^225^Ac), iodine-131 (^131^I), lutetium-177 (^177^Lu), rhenium-186 (^186^Re), rhenium-188 (^188^Re), astatine-211 (^211^At), etc. Radionuclides used in therapy should have suitable half-lives, long enough to deliver the radiation dose to the targeted organ/tissue. Additionally, they must emit α- and β-particles [9,22,23,24]. As in diagnostic applications, the nanoparticles are accumulated in the tumor via the EPR effect or due to active targeting. Their smaller size and surface chemistry affect their cellular uptake, thus they are currently being investigated for delivering chemotherapeutic agents into the tumor. The use of AuNPs in radiotherapy has increased the survival rate up to 86% compared to radiotherapy without AuNPs (20% survival rate) [25]. However, some nanoparticles present long-circulating properties, which may increase the radiation dose to the non-targeted organs, such as the spleen [3].

Various studies in the literature have reported the use of ^99m^Tc-labeled AuNPs as potential multifunctional agents and have shown interesting results, depending on their physicochemical characteristics and surface functionalization. Alberto et al. presented a new multifunctional ligand for AuNPs for direct labeling with ^99m^Tc-carbonyls consisting of a terminal thiol group as an anchor for the AuNP surface, a polyethylene glycol (PEG) linker, and 2,3-diaminopropionic acid (DAP) chelator for the [^99m^Tc][Tc(CO_3_)]^+^ core (HS-PEG-DAP). In addition, a small molecule inhibitor for the prostate specific membrane antigen (PSMA) was conjugated to the coating ligand to introduce a targeting function for prostate cancer [18]. Gómez-Oliván et al. functionalized AuNPs with hydrazinonicotinamide-GGC (HYNIC-GGC) for labeling with ^99m^Tc and c[RGDfK(C)] peptides for tumor targeting by means of spontaneous reaction of the thiol groups of cysteine [7]. Another example of functionalizing AuNPs via thiol groups of cysteine was provided by Torres–Garcia et al., who evaluated the in vitro potential of ^99m^Tc-labeled and ^177^Lu-labeled AuNPs conjugated to Tat(49–57)-Lys^3^-bombesin peptides (^99m^Tc/^177^Lu-AuNP-Tat-BN) as a plasmonic photothermal therapy and targeted radiotherapy system in PC3 prostate cancer cells [14]. 

A multifunctional radiopharmaceutical, such as the one described herein, ([^99m^Tc]Tc-AuNP may function simultaneously as both a radiodiagnostic imaging agent and a thermal ablation system (i.e., localized heating after laser irradiation) in cancer cells [26,27]. The aim of this initial research study was to functionalize the surfaces of AuNPs of two different sizes (2 nm and 20 nm, referred to as Au^(2)^NPs and Au^(20)^NPs, respectively), with two thiol ligands (referred to as L_1_H and L_2_H), which allow direct labeling with ^99m^Tc-carbonyls. These ligands contain a thiol group, which acts as an anchor for the surface of the AuNPs, as well as an [NNN] donor atom set for robust coordination of the [^99m^Tc][Tc(CO)_3_]^+^ group. The labeling procedure requires surface modification of the AuNPs with each ligand, resulting in Au^(2)^NPs-L_1_, Au^(2)^NPs-L_2_, Au^(20)^NPs-L_1_, Au^(20)^NPs-L_2_, and incubation with the semi aqua ion [^99m^Tc][Tc(H_2_O)_3_(CO)_3_]^+^ as the last step. [^99m^Tc]Tc(H_2_O)_3_(CO)_3_^+^ is an ion with three coordination sites. Its water molecules can be substituted, which means that a huge variety of mono-, bi-, and tridentate ligands can be used and radiolabeled with the tricarbonyl core [18,20]. In this study, the two tridentate thiol ligands L_1_H and L_2_H (shown in Figure 1) were used: (i) to stabilize the AuNPs by forming strong Au-S bonds [12] and (ii) to form a complex with the [^99m^Tc][Tc(CO)_3_]^+^ core via the three nitrogen atoms that acted as donor atoms [19,20].

After functionalization of AuNPs with the two thiol ligands and radiolabeling with [^99m^Tc][Tc(H_2_O)_3_(CO)_3_]^+^, in vitro stability in human serum and in cysteine and histidine solutions was examined in order to evaluate the stability of the formed radiolabeled complexes. Moreover, cytotoxicity studies using the MTT assay were carried out to evaluate the toxicity of the nanoconjugates. Finally, lipophilicity experiments were performed to determine the lipophilicity of the radiolabeled nanoparticles, while a hemolysis assay helped us examine the biocompatibility of bare and functionalized AuNPs. 

## 2. Materials and Methods

### 2.1. Chemical Reagents

^99m^Tc is a gamma emitter with a photon energy of 140 keV, which requires radiation protection precautions during handling to reduce the risk of harm. All work associated with radiolabeling procedures was conducted in a licensed radiochemistry facility, where such experiments could be safely conducted.

All reagents and solvents were of chemical grade and used without further purification. Dimethylsulfoxide (DMSO, > 99.5%) was purchased from Aldrich Chemical (St. Louis, MO, USA). Acetonitrile (CH_3_CN > 99.5%) was purchased from Carlo Erba (Val-de-Reuil, France). Trifluoroacetic acid (TFA > 99%) was purchased from Alfa Aesar (Loughborough, UK). Human serum was purchased from Sigma Aldrich (St. Louis, MO, USA). L_1_H and compound 1 were synthesized according to published procedures with slight modifications [28,29]. Nuclear magnetic resonance (NMR) spectra were recorded in DMSO-d6 on a Bruker Avance DRX 500 MHz spectrometer at room temperature. The measured chemical shifts are reported in δ(ppm) and the residual signal of the solvent was used as the internal calibration standard (CDCl_3_ ^1^H = 7.26 ppm, ^13^C = 77.16 ppm). All ^13^C NMR spectra were measured with complete proton decoupling. Data of NMR spectra were recorded as follows: s = singlet, d = doublet, t = triplet, m = multiplet, br = broad signal. The coupling constant *J* is reported in hertz (Hz). HPLC was performed using a Waters 600 Controller pump, a Waters 996 Photodiode Array detector, and a γ-RAM radioactivity detector to measure radioactive flow on a Jupiter C4 Column (150 × 4.60 mm, 5 μm, 300 Å, Phenomenex, Torrance, CA, USA). The UV detection wavelength was set at 220 nm for all experiments. ^99m^Tc was eluted as Na[^99m^Tc]TcO_4_ from a commercial ^99^Mo/^99m^Tc generator (Mallinckrodt Medical B.V.). The HPLC solvents of analytical grade were filtered through 0.22 mm membrane filters (Millipore, Milford, MA, USA). Radioactivity measurements were conducted in a dose calibrator (Capintec, Ramsey, NJ, USA). Samples for lipophilicity studies were measured on a Packard COBRA II Auto-Gamma Counter (Ramsey, MN, USA).

The 4T1 cell line was acquired from the cell bank of the Laboratory of Radiobiology, Institute of Nuclear and Radiological Sciences and Technology, Energy, and Safety, NCSR “Demokritos”, Athens, Greece. RPMI was used as medium for the cultures and was purchased from Biowest (Nuaillé, France) and the MTT reagent [3-(4,5 –dimethylthiazol-2-yl)-2,5-diphenyltetrazolium bromide] was obtained from Applichem (Darmstadt, Germany). For the optical density measurements, a LabSystems Multiskan RC Microplate Reader (Thermo Fisher Scientific, Waltham, MA, USA) was used.

The AuNPs were characterized by transmission electron microscopy (TEM, LEO 912B). The apparent hydrodynamic diameter and zeta potential (ζ) were measured by dynamic light scattering (DLS, Malvern, UK).

### 2.2. Synthesis of Gold Nanoparticles

Gold nanoparticles with 20 nm diameter (Au^(20)^) were synthesized by the modified Turkevich method [4]. For this purpose, 40 mg of gold(III) chloride trihydrate was dissolved in 100 mL of distilled water and heated under reflux in a round bottom flask. Next, 112 mg of trisodium citrate dihydrate previously dissolved in 10 mL of distilled water were rapidly added to the solution of gold(III) chloride and the mixture was further was for 15 min. After cooling down the gold nanoparticle solution, the flask was wrapped with aluminum foil and stored at 4–8 °C. The size of Au^(20)^ was checked by TEM and DLS techniques. The prepared solution of Au^(20)^ had great stability and exhibited a deep red color.

Gold nanoparticles with 2 nm diameter (Au^(2)^) were synthesized as described before [30]. A solution of chloroauric acid (78.8 mg in 10 mL of methanol) was added to the aqueous solution of 4-mercaptobenzoic acid (MBA 104.85 mg in 8 mL), the pH was adjusted to 13 using NaOH, and the solution was left at room temperature for 24 h. In the next step, to the solution of Au/MBA 86.67 mL of methanol, 246.67 mL of water and 3.33 mL of a 0.25 M sodium borohydride were added and the mixture was left for 18 h at RT. The solution color changed from pale yellow to black. In the last step, 350 mL of methanol was added to the above solution and the mixture was centrifugated at 8000× *g* for 10 min. The supernatant was removed and the precipitate was suspended in 16 mL of 10 mM ammonium acetate. The size of the synthesized Au^(2)^ was checked by TEM.

### 2.3. Synthesis of Ligands

**Synthesis of trityl intermediate 2:** A solution of quinoline-2-carbaldehyde (0.49 g, 3.13 mmol) in DCM (1 mL) was added dropwise within 5 min to a mixture of 2-(tritylthio)ethanamine 1 (0.5 g, 1.57 mmol) and NaBH(OAc)_3_ (1 g, 4.7 mmol) in DCM (4 mL) at 0 °C. The reaction was allowed to reach room temperature, at which point the brown-colored suspension turned into a deep brown slurry and was left to stir overnight. The next day, the reaction was quenched with sat. NaHCO_3_ (10 mL) at 0 °C and was stirred for 3 h, allowing it to reach room temperature. The reaction was then transferred to a funnel, diluted with CHCl_3_ (60 mL), and washed with sat. NaHCO_3_ (1 × 50 mL), water (1 × 50 mL) and brine (1 × 50 mL). The organic layer was dried over Na_2_SO_4_, filtered, and evaporated to dryness under vacuum. The crude red–brown oil (1.56 g) was purified by gravity column chromatography using CHCl_3_/AcOEt/Et_3_N 80:20:1 to afford the product (3, 725 mg, 77%) as a red foamy solid. ^1^H NMR (CDCl_3_, ppm): 8.13–8.07 (m, 2H, Quin*H*), 8.07–8.01 (m, 2H, Quin*H*), 7.81–7.65 (m, 6H, Quin*H*), 7.54–7.47 (m, 2H, Quin*H*), 7.40–7.32 (m, 6H, STrt), 7.23–7.10 (m, 9H, STrt), 3.89 (s, 4H, NCH_2_ Quin*H*), 2.68 (t, *J* = 6.7 Hz, 2H, NC*H*_2_CH_2_STrt), 2.41 (t, *J* = 6.7 Hz, 2H, NCH_2_C*H*_2_STrt). ^13^C NMR (CDCl_3_, ppm): 160.31, 147.61, 145.00, 136.38, 129.67, 129.42, 129.14, 127.89, 127.59, 127.52, 126.63, 126.22, 121.29, 66.68, 61.15, 53.67, 30.09.

**Synthesis of L_2_H:** TFA (3.4 mL, 44.8 mmol) was added dropwise within 10 min to a solution of 2 (412 mg, 0.68 mmol) and triethylsilane (0.34 mL, 2.12 mol) in DCM (10 mL) at −10 °C. The deep-red colored reaction was stirred at −10 °C for 5 min and then was allowed to reach room temperature, where it was stirred for 2 h. Volatiles were then removed under vacuum at 30 °C and the red/brown solid was dissolved in CHCl_3_ (30 mL) and washed with sat. NaHCO_3_ (1 × 25 mL), water (1 × 25 mL), and brine (1 × 25 mL). The organic layer was dried over Na_2_SO_4_, filtered, and evaporated to dryness under vacuum. The crude product (0.5 g) was purified by flash column chromatography using CHCl_3_ /AcOEt/Et_3_N 8:2:0.1 to afford L_2_H (216 mg, 88%) as a yellow solid (Figure 2). ^1^H NMR (CDCl_3_, ppm): 8.16–8.11 (m, 2H, Quin*H*), 8.09–8.02 (m, 2H, Quin*H*), 7.82–7.65 (m, 6H, Quin*H*), 7.54–7.47 (m, 2H, Quin*H*), 4.06 (s, 4H, NCH_2_ Quin*H*), 2.90 (t, *J* = 6.6 Hz, 2H, NC*H*_2_CH_2_SH), 2.71 (t, *J* = 6.6 Hz, 2H, NCH_2_C*H*_2_SH), 1.61 (br, 1H, CH_2_S*H*) ^13^C NMR (CDCl_3_, ppm):159.99, 147.66, 136.56, 129.58, 129.15, 127.65, 127.52, 126.37, 121.30, 61.29, 57.45, 22.64.

### 2.4. Functionalization of Gold Nanoparticles

Prior to the radiolabeling process, both Au^(2)^ and Au^(20)^ were each surface-functionalized with both thiol ligands (L_1_H, L_2_H) in two separate procedures, leading to four nanoparticle complexes: Au^(2)^L_1_, Au^(2)^L_2_, Au^(20)^L_1_, and Au^(20)^L_2_. All samples were prepared according to the same procedure, as follows. Briefly, 0.001 nmol of AuNPs were mixed with 0.8 μmol of each ligand (0.21 mg of L_1_H and 0.29 mg of L_2_H dissolved in DMSO) and stirred overnight. Then, the samples were centrifugated at 2300× *g* for 20 min and the supernatant was carefully removed. The purified AuNPs were ready for radiolabeling, as described below in Section 2.9 (Figure 3).

### 2.5. Cell Cultures

The 4T1 murine mammary cancer cell line was used for the biological evaluation of the synthesized compounds. Cells were grown in RPMI-1640 medium (RPMI) at a pH of 7.4, supplemented with 10% fetal bovine serum (FBS), 100 U/mL of penicillin, 2 mM glutamine, and 100 μg/mL of streptomycin. Cell cultures were maintained in flasks and were grown at 37 °C in a humidified atmosphere of 5% CO_2_ in air. Subconfluent cells were detached using a 0.25% trypsin-0.53 mM ethylenediaminetetraacetic acid (EDTA) solution, and the subcultivation ratio was 1:8–1:10.

### 2.6. Cytotoxicity Studies

Cytotoxicity evaluation of bare AuNPs (namely Au^(2)^ and Au^(20)^) and ligand-functionalized AuNPs (namely Au^(2)^L_1_, Au^(2)^L_2_, Au^(20)^L_1_, and Au^(20)^L_2_) was carried out using the MTT [3-(4,5 –dimethylthiazol-2-yl)-2,5-diphenyltetrazolium bromide] colorimetric assay. This method is based on the reduction of yellow MTT tetrazolium salt to purple formazan crystals by active cells in mitochondria and catalyzed by dehydrogenase enzymes. The formed purple formazan crystals can be solubilized and the absorbance of this purple solution can be quantified and determined by spectrophotometry at a certain wavelength. This transformation is only possible in viable cells, because dead cells are unable to metabolize MTT, thus leading to reduced absorbance. The metabolic activity of the cells is proportional to the color density of the purple formazan crystals. For these reasons, MTT assay can be used as an indicator for cell viability and cytotoxicity [31,32].

4T1 cells were seeded in a 96-well plate at a density of 10^4^ cells per well and incubated overnight at 37 °C for attachment. The following day, increasing concentrations of Au^(2)^, Au^(20)^ and ligand-functionalized Au^(2)^L_1_, Au^(2)^L_2_, Au^(20)^L_1_, and Au^(20)^L_2_ (0.0625, 0.125, 0.25, 0.5, 1 and 2 nM) were added to the exponentially growing cells in 4 replicates. The control group consisted of cells without the compound. After 24 h of incubation, the medium was replaced with MTT solution (C_MTT_ =1 mg/mL) in RPMI and was added to each well. The cells were further incubated for 4 h. After removing the MTT solution, 100 μL of isopropanol was added to each well. Following thorough mixing, the absorbance was measured at 560 nm using an ELISA reader. Finally, the mean value of the optical density (OD) of the four replicates and the percentage of the OD was calculated with the following equation:$$OD\;\%=\frac{\left(mean\;value\;of\;the\;OD\;of\;four\;replicates\right)}{\left(OD\;of\;control\right)}\times 100$$

Each assay was performed in triplicate.

### 2.7. Hemolysis Assay

The lab members participating in this study provided their consent for the use of their blood samples in these experiments. The donors underwent blood sampling following the approval of the Ethics Committee of the NCSR “Demokritos”. All experiments were carried out in accordance with relevant guidelines and regulations. According to our protocol, blood was drawn by venipuncture from two healthy donors and was centrifugated at 1000× *g* for 5 min to separate the plasma from the red blood cells (RBCs). After removing the plasma, the RBCs were washed 3 times with PBS (0.01 M, pH 7.4) free of calcium and magnesium. Solutions of bare and ligand-functionalized AuNPs were prepared at different concentrations in PBS (0.5–1–2 nM). Then, 285 μL of each sample was added to 15 μL of RBCs. As the negative control, we used 285 μL of PBS from the last centrifugation with 15 μL of RBCs, whereas 10 μL of TRITON X (10%) mixed with 275 μL PBS and 15 μL of RBCs served as the positive control in our study. Finally, all the samples were incubated at 37 °C for 3 h and were centrifugated once again at 1000× *g* for 5 min. One hundred microliters of the supernatants of all the samples were placed in 96-well plates and the OD was measured at 450 nm in a LabSystems Multiskan RC Microplate Reader [33]. The hemolysis ratio was calculated with the following equation:$$Hemolysis\;ratio\;\%=\frac{\left(OD\;of\;AuNPs-OD\;of\;negative\;control\right)}{\left(OD\;of\;positive\;control-OD\;of\;negative\;control\right)}\times 100$$

### 2.8. Preparation of the Precursor [^99m^Tc][Tc(H_2_O)_3_(CO)_3_]^+^

The labeling precursor [^99m^Tc][Tc(H_2_O)_3_(CO)_3_]^+^ was prepared as described in the literature [34]. Briefly, a vial containing 4 mg Na_2_CO_3_, 20 mg sodium tartrate and 7 mg NaBH_4_ was sealed and CO gas was purged for 2 min prior to addition of 1 mL Na[^99m^Tc]TcO_4_ eluate obtained from a ^99^Mo/^99m^Tc generator. The vial was heated at 115 °C for 30 min and, at the end of the reaction, was left to cool at room temperature. Finally, the precursor was brought to pH 6.5–7 through the addition of HCl 1 M. The formation of the precursor [^99m^ Tc][Tc(H_2_O)_3_(CO)_3_]^+^ was determined by reverse-phase HPLC(RP-HPLC) on a C18-RP column with a MeOH/0.1% TFA and H_2_O/0.1% TFA gradient over 30 min at a flow rate of 1 mL/min. The radioactivity of the precursor was measured using a dose calibrator. t_R_ HPLC (γ detector): [^99m^Tc][Tc(H_2_O)_3_(CO)_3_]^+^: 4–6 min, [^99m^Tc][TcO_4_]^–^: 3.0 min. Radiochemical yield (RCY) > 95%.

### 2.9. Radiolabeling with [^99m^Tc][Tc(H_2_O)_3_(CO)_3_]^+^

After functionalization, the AuNPs were radiolabeled with 100 μL [^99m^Tc][Tc(H_2_O)_3_(CO)_3_]^+^ (0.8–2 mCi) by incubation at 70 °C for 2 h. Radiochemical yield was determined by HPLC, applying a linear gradient system from 0% B to 100% B at 25 min, where solvent A was H_2_O/0.1% TFA and solvent B was ACN/0.1% TFA, at a flow rate of 1 mL/min. Subsequently, the samples were purified by centrifugation at 2300× *g* for 20 min (Figure 2).

### 2.10. In Vitro Stability Studies

After purification, we examined the in vitro stability of the radiolabeled nanostructures in human serum and in cysteine/histidine solutions of the formed radiolabeled complexes in order to evaluate their in vivo stability.

#### 2.10.1. Human Serum Stability

For human serum stability assessment, 50 μL of each of the radiolabeled complexes (100–300 μCi) were challenged against 450 μL human serum and the mixture samples were incubated at 37 °C for 1 h and 24 h. Afterwards, 100 μL of each mixture were treated with 200 μL ethanol at different timepoints (1 h, 24 h). The samples were centrifugated at 450× *g* for 10 min to precipitate serum proteins. The supernatants were removed and analyzed by HPLC [10].

#### 2.10.2. Cysteine–Histidine Stability

For cysteine–histidine stability assessment, 50 μL samples οf each of the radiolabeled complexes (100–300 μCi) were challenged against 450 μL οf cysteine and histidine solutions (0.02 M). Then, the samples were incubated at 37 °C for 1 h and 24 h and analyzed by HPLC [35].

### 2.11. Lipophilicity Studies

The lipophilicity of the radiolabeled complexes was determined by calculating the partition coefficient (P) with the shake-flask method. Briefly, in a centrifuge tube, 1 mL of 1-octanol and 1 mL PBS (0.01 M, pH 7.4) were mixed with 1–2 μCi of each of the radiolabeled complexes. The samples were vortexed for 1 min and the radioactivity of the aliquots (200 μL) of each phase was counted in a gamma counter. The partition coefficient was calculated according the following equation. The results were expressed as logP. The procedure was repeated three times [35].
$$P=\frac{\frac{counts}{mL}on\;the\;1-octanol\;phase\;}{counts/mL\;on\;the\;PBS\;phase}$$

## 3. Results and Discussion

### 3.1. Synthesis and Characterization of Gold Nanoparticles

The synthesized Au^(2)^ and Au^(20)^ nanoparticles were characterized by TEM and DLS methods. The results are presented in Table 1 and in Figure 4. The obtained Au^(20)^ nanoparticles were characterized by TEM and DLS methods. In the case of Au^(2)^, due to the small size, only TEM analysis was possible. The results of size and zeta potential of nanoparticles are presented in Table 1. The negative zeta potential values for naked AuNPs, as well as AuNPs decorated with L_2_H, indicate that there was no tendency for the particles to aggregate. The difference between the zeta potential of non-modified AuNPs and L_2_H additionally confirms that surface modification was achieved. DLS measurements show that Au^(20)^L_1_ were five times bigger in size after surface modification and positive zeta potential.

### 3.2. Synthesis of Ligands

To enable high-affinity ^99m^Tc labeling of AuNPs, we started this study by synthesizing two bifunctional tridentate [NNN] ligands termed L_1_H and L_2_H. Each carries a thiol moiety that can robustly conjugate with AuNPs via an Au-S bond while the [NNN] donor atom set remains exposed for ^99m^Tc coordination. The L_1_H and L_2_H chelating ligands incorporate amine and aromatic N-heterocycles donors (pyridine for L_1_H and quinoline for L_2_H), for which previous studies on the coordination chemistry of the M(CO)_3_ core has shown that they are highly effective [36,37]. L_1_H was synthesized by following a published procedure with slight modifications [28], while L_2_H was synthesized as described above (both synthetic procedures are shown in Figure 2). Cysteamine hydrochloride was reacted with triphenylmethanol to afford the S-trityl intermediate 1 [29]. Reaction of 1 with quinoline-2-carbaldehyde under reductive amination conditions yielded 2 in good yield (77%) and the trityl protecting group was subsequently cleaved with TFA/Et_3_SiH to afford L_2_H (88%).

### 3.3. MTT Assay

The MTT assay was conducted to evaluate the cytotoxicity of both bare and functionalized AuNPs against 4T1 murine mammary cancer cells. The murine 4T1 mammary epithelial carcinoma represents a widely used model of triple-negative breast cancer (TNBC) [38]. The 4T1 tumor shares many features of human stage IV breast cancer, including metastatic behavior in mice, with spontaneous metastasis to lymph nodes, lung, liver, bone, and brain while the primary tumor is still growing [39]. Furthermore, for our future prospects on assessing the therapeutic potential of nanoparticles radiolabeled with beta- or alpha-emitting radionuclides, the 4T1 cell line provides an interesting tumor model with spontaneous metastases to various sites (e.g., lung, bone, etc.) [40].

The samples were exposed to different concentrations of AuNPs for 24 h. The results of the experimental data are summarized in Figure 5 and Figure 6 and indicate that Au^(2)^ and Au^(20)^ revealed almost no toxicity, even at the highest tested concentration. In the case of the nanoparticles functionalized with L_1_H, Au^(2)^L_1_ revealed significant toxicity, with cell viability decreasing gradually from 77% at the lowest concentration (0.0625 nM) to 6% at the highest concentration (2 nM), whereas Au^(20)^L_1_ revealed similar toxicity patterns, with cell viability decreasing from 96% to 17%. On the contrary, the nanoparticles functionalized with L_2_H showed less toxicity in 4T1 cells since the recorded cell viability values ranged between 91–77% for Au^(2)^L_2_ and 87–84% for Au^(20)^L_2_ at the lowest (0.0625 nM) and the highest tested concentrations (2 nM), respectively. In addition, the cytotoxicity of the plain thiol ligands L_1_H and L_2_H was also evaluated and the results are summarized in Appendix A. In accordance with the recorded results of AuNPs, for ligand L_1_H, the recorded cell viability values gradually decreased from 47% to 7%, while in the case of L_2_H, the levels of viability fluctuated around 70% at all tested concentrations, indicating that the presence of Au^(2)^ and Au^(20)^ nanoparticles did not alter the biological effect of the ligands on the cells. It is worth mentioning that, in all cases, the presence of pyridine-bearing bifunctional ligand L_1_H on the surface of both Au^(2)^ and Au^(20)^ induced a more profound cytotoxic effect in 4T1 cells than the quinoline- bearing ligand L_2_H, a fact that may be related to the existing differences in structure, size, stereochemistry, etc., and this requires further investigation.

### 3.4. Hemolysis Assay

The hemolysis assay allowed us to evaluate the interactions between RBCs and the developed AuNPs. When a compound is toxic for RBCs, the tonicity of RBCs changes. In our case, 285 μL of PBS from the last centrifugation with 15 μL of RBCs was used as the negative control, whereas 10 μL TRITON X 10% (which is known to cause hemolysis) with 275 μL PBS and 15 μL of RBCs was used as the positive control. The effect of hemolysis caused by AuNPs is shown in Figure 7. The results after incubation at all variable concentrations of AuNPs indicated that our samples did not cause hemolysis, which confirms the biocompatibility of both bare and functionalized AuNPs.

### 3.5. Functionalization and Radiolabeling of AuNPs

The functionalization of Au^(2)^ and Au^(20)^ with L_1_H and L_2_H and the consequent radiolabeling procedures have been described above. The aim of the functionalization was to provide an adequate chelator for the robust radiolabeling of Au^(2)^ and Au^(20)^ with [^99m^Tc][Tc(H_2_O)_3_(CO)_3_]^+^. After the functionalization, strong Au-S bonds were formed between the AuNPs and the thiol groups of both ligands. Thiol groups are considered to be the most important type of molecule to stabilize any size of AuNPs [18,20,41]. For radiolabeling, a mixture of [^99m^Tc][Tc(H_2_O)_3_(CO)_3_]^+^ and functionalized AuNPs was incubated at 70 °C for 2 h. The semi aqua ion [^99m^Tc][Tc(H_2_O)_3_(CO)_3_]^+^ has three available coordination sites and the metal center is primarily found in the oxidative state +1, so after radiolabeling, the water molecules were substituted with the three nitrogen atoms (as donor atoms) of the thiol ligands, thus forming a ^99m^Tc complex (Figure 3). Radiolabeling yield was found to be approximately 85%. After purification by centrifugation, radiochemical purity for all samples was >95%, which remained stable for at least 24 h post-purification. The results are summarized in Figure 8. All the corresponding HPLC traces can be seen in the Appendix A.

### 3.6. In Vitro Stability Studies

The radiolabeled complexes exhibited satisfactory in vitro stability in human serum, and negligible transchelation in the presence of cysteine and histidine, which indicates the robust binding of ^99m^Tc with the AuNPs, thus enhancing their potential as SPECT imaging agents or for use in in vivo tracking applications. Our results are in accordance with work performed by other groups [35,42]. The results are summarized in Figure 9 The corresponding HPLC traces can be seen in the Appendix A.

### 3.7. Lipophilicity Studies—Determination of Partition Coefficient

The partition coefficients (P) were determined using the shake-flask method as described above and are expressed as logP. The AuNPs functionalized with L_1_ exhibited hydrophilic behavior, whereas the AuNPs functionalized with L_2_, indicated higher lipophilicity, which may be attributed to the more lipophilic character of L_2_H due to the quinoline ring. The results are presented in Table 2.

## 4. Conclusions

Au^(2)^ and Au^(20)^ nanoparticles were surface-functionalized with two different [NNN]-bearing ligands capable of coordinating the [^99m^Tc][Tc(CO)_3_]^+^ core, which also contains a thiol group that acts as an anchor for the surface of the AuNPs. Radiolabeling was quick and efficient, with the resulting nanoconstructs exhibiting negligible transchelation of the radiolabel, as demonstrated by the cysteine/histidine challenge.

While robust radiolabeling was afforded for both nanoparticles functionalized with the thiol-bearing ligands L_1_H and L_2_H, further investigation on the in vivo kinetics of Au^(2)^ and Au^(20)^ nanoparticles will be performed with the L_2_-functionalized AuNPs, due to their better in vitro toxicity profile.

## Figures and Tables

**Figure 1 nanomaterials-11-02406-f001:**
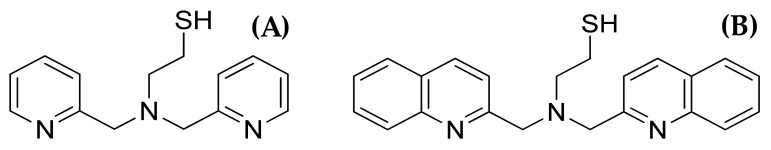
The two tridentate thiol ligands used in this study: (**A**) L_1_H and (**B**) L_2_H.

**Figure 2 nanomaterials-11-02406-f002:**
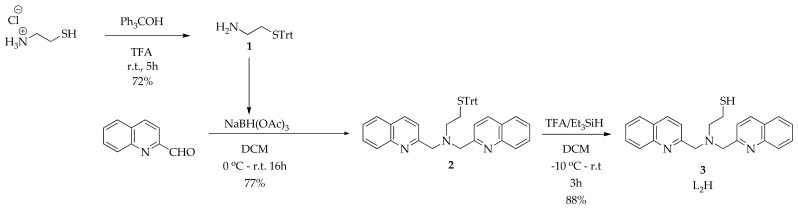
Synthesis of bifunctional ligand L_2_H. The synthesis of L_1_H has been previously reported in the literature [28,29].

**Figure 3 nanomaterials-11-02406-f003:**
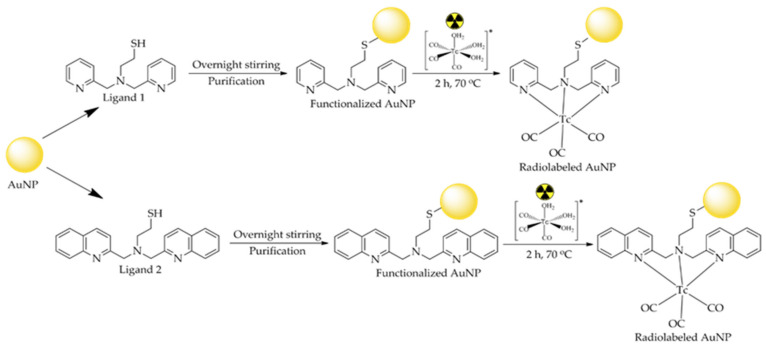
Functionalization of AuNPs with thiol ligands and radiolabeling with [^99m^Tc][Tc(H_2_O)_3_(CO)_3_]^+^.

**Figure 4 nanomaterials-11-02406-f004:**
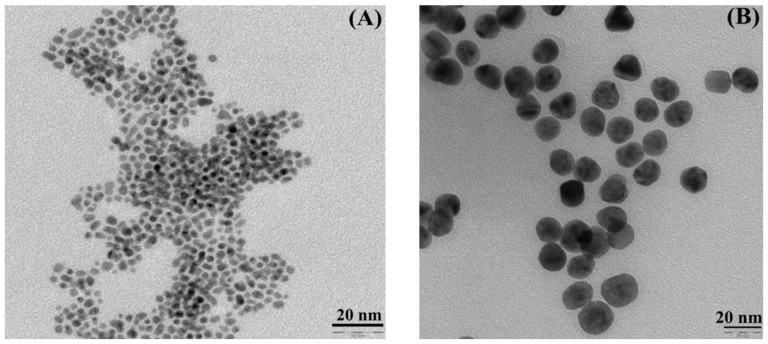
TEM images of (**A**) Au^(2)^ and (**B**) Au^(20)^ nanoparticles before surface functionalization.

**Figure 5 nanomaterials-11-02406-f005:**
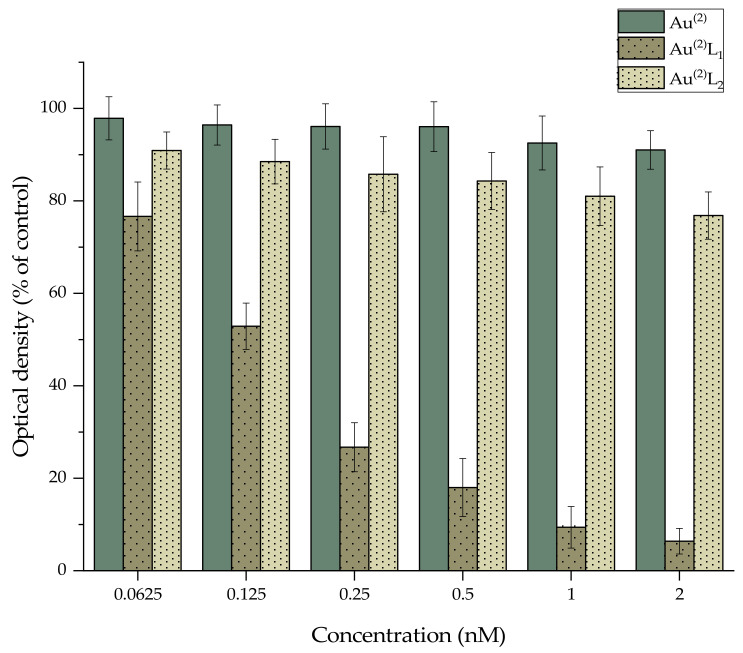
MTT assay of 4T1 cells treated with different concentrations of Au^(2)^, Au^(2)^L1, and Au^(2)^L2.

**Figure 6 nanomaterials-11-02406-f006:**
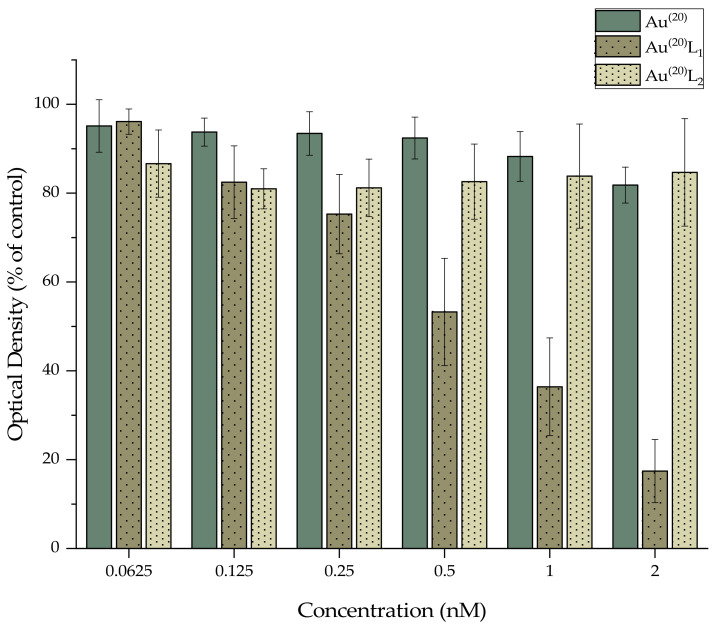
MTT assay of 4T1 cells treated with different concentrations of Au^(20)^, Au^(20)^L1 and Au^(20)^L2.

**Figure 7 nanomaterials-11-02406-f007:**
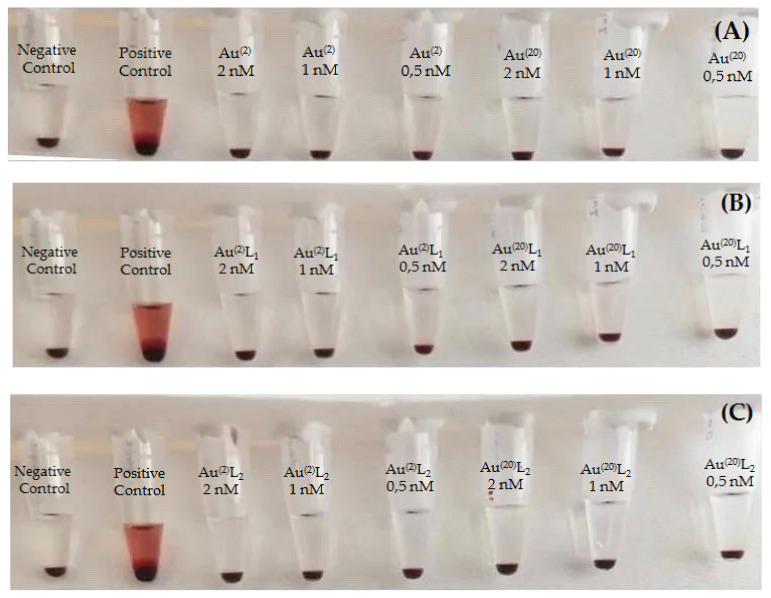
Hemolytic effect of: (**A**) Au^(2)^ and Au^(20)^, (**B**) Au^(2)^L1 and Au^(20)^L1, and (**C**) Au^(2)^L2 and Au^(20)^L2 at different tested concentrations (0.5 nM, 1 nM, and 2 nM). Positive control: 10 μL TRITON X 10% + 275 μL PBS + 15 μL of RBCs; negative control: 285 μL PBS +15 μL of RBCs.

**Figure 8 nanomaterials-11-02406-f008:**
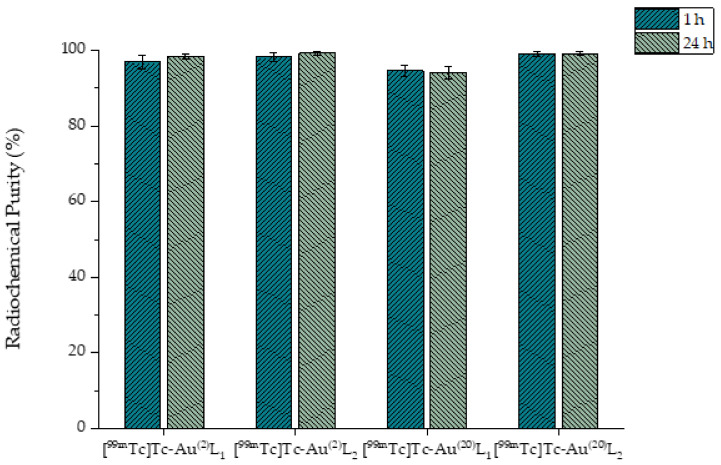
HPLC results representing the radiochemical purity of [^99m^Tc]Tc-Au^(2)^L1, [^99m^Tc]Tc-Au^(2)^L2, [^99m^Tc]Tc-Au^(20)^L1, and [^99m^Tc]Tc-Au^(20)^L2 at 1 h and 24 h post-radiolabeling.

**Figure 9 nanomaterials-11-02406-f009:**
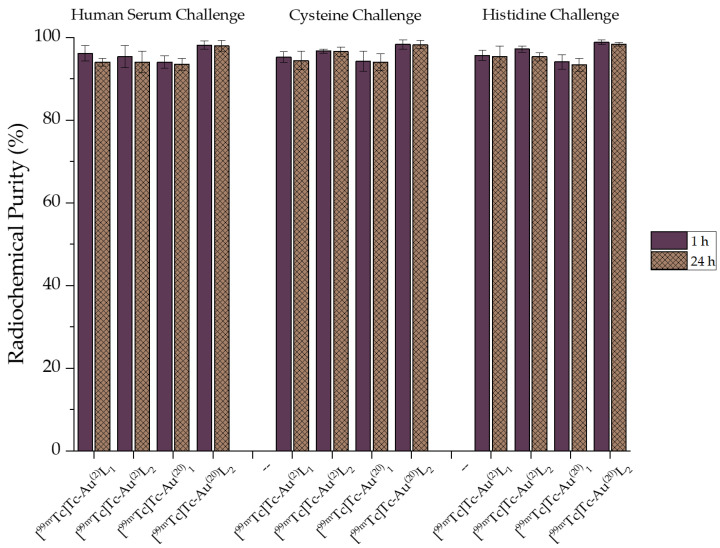
In vitro stability of [^99m^Tc]Tc-Au^(2)^L1, [^99m^Tc]Tc-Au^(2)^L2, [^99m^Tc]Tc-Au^(20)^L1, and [^99m^Tc]Tc-Au^(20)^L2 at 1 h and 24 h post-radiolabeling.

**Table 1 nanomaterials-11-02406-t001:** Hydrodynamic diameters and zeta potentials of 20 nm naked AuNPs and ligand-bearing AuNPs.

	Hydrodynamic Diameter (nm)
Au^(20)^	20.9
Au^(20)^L_1_	124.48
Au^(20)^L_2_	41.2
	**Zeta Potential (mV)**
Au^(20)^	−33.5
Au^(20)^L_1_	12.2
Au^(20)^L_2_	−16.8

**Table 2 nanomaterials-11-02406-t002:** Partition Coefficient of [^99m^Tc]Tc-Au^(2)^L1, [^99m^Tc]Tc-Au^(2)^L2, [^99m^Tc]Tc-Au^(20)^L1 and [^99m^Tc]Tc-Au^(20)^L2.

Radiolabeled Complex	logP
[^99m^Tc]Tc-Au^(2)^L1	−0.94 ± 0.24
[^99m^Tc]Tc-Au^(20)^L1	−0.85 ± 0.16
[^99m^Tc]Tc-Au^(2)^L2	2.39 ± 0.17
[^99m^Tc]Tc-Au^(20)^L2	2.23 ± 0.20

## Data Availability

The data presented in this study are available on request from the corresponding author.

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
