# Peer review of "Synthesis and In Vitro Evaluation of Gold Nanoparticles Functionalized with Thiol Ligands for Robust Radiolabeling with 99mTc"

_nanomaterials, 2021, doi:10.3390/nano11092406_

Round 1
Reviewer 1 Report
- The introduction part is not full, some closely related references can be read and cited (doi.org/10.1016/j.msec.2017.04.052 and doi.org/10.1039/C2NR32405A). and should compare the differences between them.
- The scale bars in TEM images are not clear. In addition, the TEM images after surface modification are missing.
Author Response
Please find our rebuttal in the attached file.

Reviewer 2 Report
The goal of the present study was to prepare AuNPs (2 nm and 20 nm AuNPs) functionalized with two bifunctional thiol ligands (L1 aand L2). Subsequently, the AuNPs were labeled with 99mTC, and the 99TmTC-labeled AuNPs may apply in vivo imaging or tracking of drug-carrying AuNPs. The authors pay their efforts to characterize the physico-chemical properties and cytotoxicity of L1- and L2-labeled AuNPs. Moreover, they also investigated the stability of 99TmTC-labeled AuNPs in serum, cystein-containing solution, or histidine-containing solution. Although some interesting data are presented in this study, several points should be clarified further.
- The authors use human RBCs and serum in this study. It should have IRB approval.
- Since the authors suggest that 99TmTC-labeled AuNP can be used for in vivo imaging or tracking drug-loaded AuNP, the author should compare the utility of 2 nm and 20 nm AuNP for this purpose.
- The authors should explain why L1 causes the hydrodynamic diameter and Zata potential of AuNPs to increase. In addition, the TEM of AuNP labeled with L1 and L2 should also be included.
- The authors should further clarify the importance of analyzing the cytotoxicity of L1 and L2 labeled AuNPs to 4T1 cells. Moreover, the authors should analyze the cytotoxicity of L1- and L2-labeled AuNPs on different cells including human cancer cells.
- The authors should analyze the effect of L1 and L2 on the viability of 4T1 cells.
- Since L1-labeled AuNPs only show cytotoxicity to 4T1 cells but not to RBCs, the authors should further clarify the possible cytotoxic mechanism.
- The authors suggest that the cytotoxicity of L1-labeled AuNPs is due to MMP inhibition (lines 407-411). To prove this proposition, the authors should provide the experimental evidence that MMP is important for the survival of 4T1 cells.
- The authors should analyze the cytotoxicity of 99TmTC-labeled AuNPs on 4T1 cells and RBCs.
- The authors should clarify whether the lipophilicity properties of 99TmTC-labeled AuNPs is important for its application in in vivo imaging or tracking of drug-carrying AuNPs.
Minor
- 4 and Fig.5 do not provided useful information for this study. Therefore, the two figures can be deleted.
- Some mistakes on Table 2, for example “-0,94 ± 0,24” should be “-0.94 ±24”.
Author Response
Please find a point-to-point rebuttal in the attached file.

Reviewer 3 Report
The authors present work on the synthesis and In Vitro Evaluation of Gold Nanoparticles Functionalized with Thiol Ligands for Robust Radiolabeling with 99mTc. The topic is of interest. The manuscript is well written and organized. Some minor changes are needed but in my opinion it is publishable in the Nanomaterials journal. My minor remarks are given below:
- Table 1. Why the results of the hydrodynamic diameter are not given for Au2 samples?
- It is possible to determine the quantity of adsorbed species on the gold surface?
- The adsorbed thiols stay on the surface after tests? Or some leaching is present?
- Figure 9. More information in the caption should be given. What is the difference between A, B and C?
Author Response

(The authors gave the same response as above.)

Round 2
Reviewer 2 Report
The revised manuscript addressed the most concerned issues in the reviewers’ comments. Since Au(2)L1 and Au(20)L1 show the cytotoxicity to 4T1, the authors should discuss whether Au(2)L1 and Au(20)L1 can be used for in vivo imaging or tracking drug-loaded AuNP.
